# Vector Surveillance, Host Species Richness, and Demographic Factors as West Nile Disease Risk Indicators

**DOI:** 10.3390/v13050934

**Published:** 2021-05-18

**Authors:** John M. Humphreys, Katherine I. Young, Lee W. Cohnstaedt, Kathryn A. Hanley, Debra P. C. Peters

**Affiliations:** 1Pest Management Research Unit, Agricultural Research Service, US Department of Agriculture, Sidney, MT 59270, USA; 2Jornada Experimental Range Unit, Agricultural Research Service, US Department of Agriculture, Las Cruces, NM 88003, USA; kiy761@nmsu.edu (K.I.Y.); deb.peters@usda.gov (D.P.C.P.); 3Department of Biology, New Mexico State University, Las Cruces, NM 88003, USA; khanley@nmsu.edu; 4Arthropod-Borne Animal Disease Research Unit, Agricultural Research Service, US Department of Agriculture, Manhattan, KS 66502, USA; lee.cohnstaedt@usda.gov

**Keywords:** West Nile, mosquito, vector, spatiotemporal, neuroinvasive disease, Bayesian

## Abstract

West Nile virus (WNV) is the most common arthropod-borne virus (arbovirus) in the United States (US) and is the leading cause of viral encephalitis in the country. The virus has affected tens of thousands of US persons total since its 1999 North America introduction, with thousands of new infections reported annually. Approximately 1% of humans infected with WNV acquire neuroinvasive West Nile Disease (WND) with severe encephalitis and risk of death. Research describing WNV ecology is needed to improve public health surveillance, monitoring, and risk assessment. We applied Bayesian joint-spatiotemporal modeling to assess the association of vector surveillance data, host species richness, and a variety of other environmental and socioeconomic disease risk factors with neuroinvasive WND throughout the conterminous US. Our research revealed that an aging human population was the strongest disease indicator, but climatic and vector-host biotic interactions were also significant in determining risk of neuroinvasive WND. Our analysis also identified a geographic region of disproportionately high neuroinvasive WND disease risk that parallels the Continental Divide, and extends southward from the US–Canada border in the states of Montana, North Dakota, and Wisconsin to the US–Mexico border in western Texas. Our results aid in unraveling complex WNV ecology and can be applied to prioritize disease surveillance locations and risk assessment.

## 1. Introduction

West Nile virus (WNV) is the most common arthropod borne virus (arbovirus) in the United States (US) and is the leading cause of viral encephalitis in the country [1]. Since the first US identification in New York of 1999 [2], WNV has spread from Canada to South America, and autochthonous transmission has occurred in every state in the continental US [1,3]. During this period, West Nile Disease (WND) has affected tens of thousands of US persons, with thousands of new cases continuing to be reported annually [4]. WND is clinically described as either non-neuroinvasive or neuroinvasive, with non-neuroinvasive being characterized by relatively minor symptoms (headache, myalgias, and gastrointestinal discomfort) and neuroinvasive disease presenting with aseptic meningitis, encephalitis, or acute flaccid paralysis [5]. Although the majority of persons infected with WNV are asymptomatic, about 25% of those infected experience flu-like symptoms, and approximately 1% of these cases progress to acute neuroinvasive disease with severe encephalitis and a risk of death [6,7,8]. Due to the high incidence of asymptomatic cases and uneven testing across the US, tracking and forecasting WNV occurrence and WND spatial and temporal spread has been problematic [9].

WNV transmission is shaped by a variety of extrinsic (environmental) and intrinsic (physiological) drivers, including climate, land use, vector and host competence, host immunity, and behavioral interactions between individual organisms (e.g., contact processes between a vector and host), and ecological community characteristics (e.g., species composition and abundance) [3,10]. The WNV enzootic cycle has two main components, mosquito vectors and avian hosts that amplify WNV. Incorporating these two components into statistical models for the purpose of disease forecasting is complicated by fluctuating intra- and inter-annual climate conditions, shifting vector and host abundances, and community interactions that alter WNV transmission through time and across geographic space [11,12,13,14]. As with the enzootic cycle, human vulnerability to WND is also shaped by both intrinsic factors (e.g., acquired immunity, age) and extrinsic influences like behavior (disease exposure) and economic position [15,16,17] further challenging modeling efforts [9,18].

Given the lack of a viable vaccine to protect against neuroinvasive disease [19], research elucidating WND drivers is greatly needed to improve public health surveillance, monitoring, and risk assessment. Analytical approaches that adopt a systems perspective, follow process-based assumptions, and consider interacting biotic and abiotic components are especially promising and may offer new insights into arbovirus and disease ecology [20,21,22]. We applied Bayesian spatiotemporal modeling and a landscape perspective to assess the association of vector surveillance, host species richness, and a variety of other environmental and socioeconomic disease risk factors with neuroinvasive WND throughout the conterminous US. Our primary objective was to quantify relationships between disease incidence and extrinsic and intrinsic risk factors to better anticipate future outbreaks. We hypothesized that inclusion of mixed data representing both biotic and abiotic aspects of the enzootic cycle as well as socioeconomic variables reflecting human demographic characteristics would improve predictive power over models constructed using only environmental criteria.

## 2. Materials and Methods

### 2.1. Study Area and Data Sources

The geographic domain for our study included the contiguous US with an areal extent in excess of 9.8 million km${}^{2}$. This spatial extent is inclusive of 48 states and 3109 counties. Counties were chosen as the spatial unit for analysis because human WND incidence data (described further below) were reported at the county level. Similarly, the time period of analysis (2004 to 2018) was based on the WND data period of availability. We focused on the May to August summer season, which is recognized as the peak time for WND transmission [23,24].

Response data: Human WND incidence data were obtained from the [4] as text files without any accompanying personal identifying information. Tabulated data provided the annual number of confirmed neuroinvasive and non-neuroinvasive WND cases reported within each county.

Driver data: In addition to Human WND data used as response variables, the number of annually reported avian and mosquito WNV detections resulting from surveillance were also obtained from the Centers for Disease Control and Prevention on 21 Aug 2019 [4]. Human population and economic data were acquired from the US Census Bureau as text files (https://www.census.gov/) on 21 Aug 2019. Broad age-class groupings allowed for the proportion of persons over 54 years of age in each county to be approximated; these individuals are at elevated neuroinvasive WND risk [7,8,16]. The Small Area Income and Poverty Estimates (SAIPE) Program within the Census Bureau provides annual economic statistics by county, including Median Household Income estimates used in this study [25]. The Census Bureau Topologically Integrated Geographic Encoding and Referencing database (TIGER) was also queried to obtain locations as geographic polygons depicting the areal extent of American Indian/Alaska Native/Native Hawaiian Areas (AIANNH) (https://tigerweb.geo.census.gov/tigerwebmain/TIGERweb_main.html). Last accessed on 21 Aug 2019. The AIANNH data sets were incorporated to account for documented healthcare and disease reporting inequalities by population [26,27].

Avian species occurrence data were downloaded from the Cornell Lab of Ornithology eBird database [28]. Species selected for analysis included the 100 most WNV competent species of the Passeriformes (perching birds) in the US as identified and ranked by [14]. Point-level species observations provided by eBird were geographically cross-referenced to the US county of occurrence to determine the number of unique and competent avian species (Competent Host Richness) by county for the summer season. The Competent Host Richness variable is depicted in the Appendix (Figure A1).

Climate data documenting mean maximum temperature and total precipitation for the summer season in each year were obtained from the PRISM Climate Group at a 4 km grid resolution [29] and then aggregated to the county level.

A synthetic variable was then created that reflected the recency and intensity of past WND in each county. This Historic Prevalence variable, was calculated using a rolling average of historic disease incidence (disease Cases/100,000 people) for each county. After calculating the rolling average, each year in our database was matched to the rolling average disease incidence estimated for the preceding year. The resulting synthetic variable was then scaled (0–1) where a value of 1 (one) indicated recent or recurrent high disease incidence and values near 0 (zero) were indicative of distant (historic past) outbreaks or persistent but relatively low incidence rates. The Historic Prevalence variable was intended to approximate within county, per capita temporal changes in underlying population immunity [13,30,31,32] in a way conceptually similar to measures like Force of Infection [33,34,35,36]. However, lacking detailed data describing the age and infection status of individuals, Historic Prevalence was calculated using estimates for total population and past outbreaks [37].

Data standardization: All data were geographically and temporally cross-referenced to corresponding county and year to produce a database summarizing Median Household Income, the human population Proportion ≥54 Years, avian Competent Host Richness, mean Maximum Temperature, mean Total Precipitation, the count of WNV Mosquito Detections, the count of WNV Avian Detections, estimated human AIANNH Population density, the areal extent of AIANNH lands, and total county Geographic Area. Geographic Area (km${}^{2}$) was included to quantify variation due differences in individual county sizes (areal extents or sample unit sizes) and is a common variable applied in ecological and spatial modeling [38,39,40,41].

All variables were scaled to one standard deviation and centered on the mean to facilitate later interpretation. To avoid potential multicollinearity between variables, we applied collinearity diagnostics for independent variables [42] using the perturb r-package [43]. Evaluating variables produced a condition index score of 13.60 with individual variable decomposition proportions contributing less than 0.50. These scores fell well below the diagnostic threshold of 30.00 proposed by [42], suggesting a low risk of multicollinearity during modeling.

### 2.2. Statistical Model

We constructed Bayesian spatiotemporal models to determine the association of vector surveillance, host species richness, and other environmental factors to neuroinvasive WND risk in the conterminous US through time. To improve neuroinvasive risk estimates, a joint modeling approach was employed that supported concurrent estimation of both the non-neuroinvasive and neuroinvasive diseases. For simplicity, the non-neuroinvasive and neuroinvasive disease presentations are hereafter referred to as “diseases” when referenced in combination. Modeling the two diseases simultaneously enabled us to identify the spatiotemporal distributions attributable to each disease individually as well as the spatiotemporal patterns common to both. Because the diseases share many common environmental and vector risk factors, we leveraged estimates for non-neuroinvasive disease to improve estimates for neuroinvasive disease, which is the focus of our analysis. From a statistical perspective, joint spatial disease modeling allowed information to be shared between different diseases and from locations in close proximity, thereby, reducing disparity between surveillance data sources and improving risk estimation [44,45,46].

Although our two-tier, joint model included both disease-specific and shared statistical terms, each model tier incorporated a Poisson distribution as follows,
$$\begin{array}{cc} O_{\mathrm{st}}^{d}|\theta_{\mathrm{st}}^{d} & ∼\mathrm{Poisson}(E_{\mathrm{st}}^{d},\theta_{\mathrm{st}}^{d}),\;d=1,2 \end{array}$$
where $O_{\mathrm{st}}^{d}$ signifies the disease case count observed (*O*) for each disease (*d*) in US County *s*
(s=1,2,3,…,3109) during year *t*
(t=2004,2005,2006,…,2018) as conditioned on the relative risk (θ) at that time and location. Reported case counts for non-neuroinvasive (*d* = 1) and neuroinvasive (*d* = 2) diseases were provided by the CDC as non-negative integers with no accompanying demographic information detailing age or gender groups. Although direct or indirect standardization by risk group is preferred [47], the lack of structured demographic information necessitated that expected counts (*E*) be calculated by multiplying disease-specific average rates for the period of record (2004–2018) by the population in each county (*s*) and year (*t*). Note that ${\hat{\theta }}_{\mathrm{st}}=O_{\mathrm{st}}/E_{\mathrm{st}}$ corresponds to the Standardized Incidence Ratio (SIR), which although frequently used in epidemiological analysis can be problematic to interpret due to high variance between observed and expected case counts. Our Bayesian spatiotemporal implementation offers one solution to resolving this issue [44,48].

The joint model’s process components (linear predictors) were customized to provide disease-specific covariate coefficient estimates while accounting for latencies (e.g., unmeasured or unmodeled risk factors, spatial correlation, extra-Poisson variation) within and between disease (*d*) distributions. The process components were specified such that,
(1)$$\begin{array}{cc} & \log(\theta_{\mathrm{st}}^{1})=\\ \;\alpha^{1}+\beta^{1}\cdot Z_{st}^{1}+\zeta_{s}^{1}+\gamma_{t}^{1}+\delta_{\mathrm{st}}^{1}+\lambda S_{s}, \end{array}$$
(2)$$\begin{array}{cc} & \log(\theta_{\mathrm{st}}^{2})=\\ \;\alpha^{2}+\beta^{2}\cdot Z_{st}^{2}+\zeta_{s}^{2}+\gamma_{t}^{2}+\delta_{\mathrm{st}}^{2}+\frac{1}{\lambda }S_{s}, \end{array}$$
where $\alpha^{1}$ and $\alpha^{2}$ are intercepts, respectively, giving mean risk for non-neuroinvasive (Equation (Equation 1)) and neuroinvasive (Equation (2)) disease with $\beta^{d}$$\left(\beta^{d}=\beta_{1}^{d},\ldots ,\beta_{x}^{d}\right)$ terms providing disease-specific coefficient vectors corresponding to covariate matrices ($Z_{st}^{x}$) encoding the vector surveillance, host species richness, and other environmental variables described in Section 2.1. The model included tier-specific random effects to account for spatial structure ($\zeta_{s}$), temporal structure ($\gamma_{t}$), and the space-time interaction ($\delta_{\mathrm{st}}$) exhibited by each disease, as well as a shared spatial component ($S_{s}$) to quantify interaction between diseases.

Following [49,50], spatial effects were incorporated using a modified and scaled version of the Besag–York–Mollié (BYM) model. In comparison to the classic BYM model [51], the scaled and modified BYM reduces confounding [52] between spatially structured and unstructured model components and helps facilitate interpretation of hyperparameters [53]. Specifically, disease-specific spatial effects ($\zeta_{s}^{d}$) in the joint model had the parameterization,
(3)$$\begin{array}{c} \zeta_{s}^{d}=\frac{1}{{\sqrt{\tau }}_{\zeta }}\left(\sqrt{1-\phi v}+\sqrt{\phi u_{*}}\right) \end{array}$$
(4)$$\begin{array}{c} \mathrm{Var}\left(\zeta_{s}^{d}|\tau_{\zeta },\phi \right)=\tau_{\zeta }^{-1}\left(\left(1-\phi \right)I+\phi Q_{*}^{-}\right), \end{array}$$
where $u_{*}$ (Equation (Equation 3)) is a scaled, spatially structured component (Gaussian Markov random field) in which US counties are considered conditionally independent unless adjoining as neighbors (sharing a common geographic boundary point), $Q_{*}$ is a precision matrix with scaled, generalized inverse $Q_{*}^{-}$ (Equation (Equation 4)) derived from an adjacency matrix (“neighborhood graph”) that identifies neighboring counties, and v is an unstructured component to account for overdispersion not attributable to spatial structure [53]. The county-based neighborhood graph was constructed using the spdep r-package [54] with a neighbor contiguity parameter set to consider counties sharing a common boundary point (i.e., a “queen” configuration) as adjacent. Both the spatially structured ($u_{*}$) and unstructured (v) components were standardized to exhibit a variance of one with marginal precision denoted as τ. The proportion of marginal variance attributable to spatial structure is given as ϕ, which falls within the range 0≤ϕ≤1 and implies only spatially structure at the value ϕ=1. This formulation results in the covariance matrix described by Equation (Equation 4). The shared spatial component ($S_{s}$, Equations (Equation 1) and (2)) was likewise implemented as a scaled and modified BYM as detailed for the disease-specific spatial terms ($\zeta_{s}^{d}$), except that risks were weighted by a parameter (λ, Equations (Equation 1) and (2)) to allow the shared effect to vary between model tiers. In epidemiological terms, although the non-neuroinvasive and neuroinvasive disease distributions were anticipated to be geographically correlated to some degree (due to common risk factors), variation in surveillance, reporting, and disease-specific risks often result in distributions that differ in space (i.e., there may not be 100% correspondence). The model estimated, scaling parameter λ quantifies the degree of spatial correspondence or “interaction” between the two diseases.

Correlated time ($\gamma_{t}^{d}$, Equations (Equation 1) and (2)) was modeled using a dynamic order 1 random walk (RW1) defined as $\gamma_{t}^{d}=\gamma_{t-1}^{d}+\Delta \gamma_{t}^{d}$, such that the value at the current time step was based on the prior step plus an incremental value $\Delta \gamma_{t}^{d}$, where $\Delta \gamma_{t}^{d}=N\left(0,\sigma^{2}\right)$ and sums to zero. Disease-specific, space-time interaction terms ($\delta_{\mathrm{st}}^{d}$, Equations (Equation 1) and (2)) were specified as independent and identically distributed random effects (IID) with variable groups consisting of unique county-year combinations. The dynamic random walk effect aided in capturing temporal trends across the period of record (2004–2018), whereas, space-time interaction helped identify those locations subject to above or below average disease risk with respect to the period of record mean.

Spatiotemporal models can be computationally demanding to run, therefore, we opted to use Integrated Laplace Approximation as a more efficient yet fully Bayesian alternative to Markov chain Monte Carlo methods [55,56,57]. Spatial and temporal effects were specified with weakly informative Penalizing Complexity priors [50,53] having parameters $p_{1}=1$ and $p_{2}=0.001$ with enforced zero mean constraints to help reduce confounding between covariates. Fixed effects were assigned vague zero mean normal priors with a 0.0001 precision.

A comparative modeling approach was adopted to identify the model formulation that exhibited greatest parsimony. Five different models were constructed and compared before performing model selection. First, a relatively simple regression model was fit (Model1) that included all vector surveillance, host species richness, and other fixed covariates of interest but without any random effects to account for spatial and temporal variability. In contrast to Model1, a second model (Model2) included all spatiotemporal effects, but, excluded all fixed covariates. Comparison of Models 1 and 2 aided in verifying the need for inclusion of spatiotemporal effects. Building from Models 1 and 2, a single-tiered model (Model3) incorporated all fixed and random effects used in estimating neuroinvasive disease, except that it was designed as a single-tier model and was not jointly fit with non-neuroinvasive disease cases. Model3 helped gauge the benefit of fitting both disease presentations jointly. Model4 was constructed to jointly estimate both diseases, but only incorporated spatiotemporal effects in the absence of fixed covariates. Finally, a full model (Model5) was constructed that comprised both fixed and random effects (Equations (Equation 1) and (2)) and jointly estimated both diseases concurrently.

The Watanabe–Akaike information criterion (WAIC) and deviance information criterion (DIC) were used for model comparison and selection. Both the WAIC and DIC indicate the relative fit of each model given the available data and are scaled such that the lower the value, the better the model. Although our study focused on assessing the relative importance of several biotic, abiotic, and demographic variables rather than forecasting future disease occurrence, we nonetheless opted to perform model validation. Validation was undertaken by first removing year 2018 observations from our data set, training the full joint model (Model5) with 2004–2017 data, and then predicting year 2018 disease occurrences out-of-sample. At the time of analysis, reported WND case counts had not been finalized by the CDC. Formal predictions for 2018 were then compared to observed disease case counts using logarithmic [58] and Brier scores [59]. Specifically, model predicted exceedance probabilities for 1, 2, 5, and 10 neuroinvasive cases were compared to county-specific case counts reported in 2018.

## 3. Results

The full, joint disease model that included both fixed and random effects exhibited the best overall parsimony (Table 1). Comparison of parsimony metrics also indicated that spatiotemporal random effects (Model2) explain more of the observed disease variation than environmental covariates alone (Model1). Combining both spatiotemporal and fixed effects into a single model led to improved parsimony (Model3) over models that evaluated either covariate types separately.

Estimated coefficients imply that as household income, Historic Prevalence, competent avian host richness, and AIANNH populations increase, disease risk lessens, whereas warmer temperatures, older populations, larger county areas, and increased WNV detections in mosquitoes and birds all contribute to elevated neuroinvasive WND risk. Model estimated covariate coefficients from the best performing model (Model5) are listed in Table 2 for West Nile neuroinvasive disease. Note that with the exception of total precipitation, all covariates were important predictors of neuroinvasive disease as judged by 95% credible intervals (95% CI) excluding zero. Exponentiation of the intercept −0.54(−0.86, −0.22 95% CI) indicates that the mean rate of neuroinvasive disease for the period of record was approximately 0.58(0.42, 0.8 95% CI) Cases/100,000 people. Subsequent inspection of fitted model values revealed the median rate to be about 0.46 Cases/100,000 people. Table 2 coefficients are reported on the log-scale and can be interpreted with respect to a percent (%) change in median disease case count for each covariate unit increase while holding all other covariates constant (i.e., keeping covariates at their mean value). For example, the negative coefficient for Median Household Income was approximately −0.08(−0.12, −0.04 95% CI), which translates to about a 8.33% [(exp(0.08) − 1) × 100%] decrease in the median rate for each standard deviation (sd ∼ $12,271) increase in Median Household Income. By comparison, each standard deviation (sd ∼ 6.41%) increase in the proportion of a county’s population over 54 years raises median disease cases by 305.52% [(exp(1.40) − 1) × 100%] holding all other covariates constant. This rate is consistent with other studies finding dramatic rate increases associated with older populations [7,8,16] and is discussed further in Section 4. The interaction term λ (Equations (Equation 1) and (2)) quantifies the spatiotemporal relationship between non-neuroinvasive and neuroinvasive disease. The Disease Interaction estimate of 0.89 (0.20, 0.04 95% CI) indicates that non-neuroinvasive WND risk was positively associated with neuroinvasive WND risk. As non-neuroinvasive risk increases, so too does the risk of neuroinvasive WND. It is also the case that the Disease Interaction credible interval excludes zero, meaning that in addition to improving model parsimony (Table 1), non-neuroinvasive disease occurrence was an important and statistically significant indicator of neuroinvasive WND risk. To avoid confusion and simplify interpretation of neuroinvasive WND risk factors, estimated coefficients specific to non-neuroinvasive disease are provided in the Appendix (Table A1).

Figure 1 compares neuroinvasive disease Standardized Incidence Rates (SIR) to Relative Risk for the period of record, Figure 2 maps mean Relative Risk and estimated disease case counts, and Figure 3 shows exceedance probabilities for 1, 2, 5, and 10 cases. Note that patterns of elevated Relative Risk illustrated in Figure 2 do not fully coincide with locations that experience the highest case counts. Although southern California and Arizona are subject to the greatest total number of neuroinvasive disease cases, northern portions of the Western US exhibit the highest-level of Relative Risk. These patterns imply that populations in Wyoming, Montana, the Dakotas, and surrounding western states experience disproportionately higher rates of neuroinvasive disease than is expected based on population size alone. To provide better historic perspective to Figure 2 and Figure 3, which represent anticipated averages given the totality of historic disease patterns and model covariates, Figure 4 illustrates the proportion of counties in each state that have documented case counts between 2004 and 2018. Values shown in Figure 4 are the smoothed estimates (i.e., fitted values) from the joint spatiotemporal model (Model5).

Out-of-sample predicted exceedance probabilities for 1, 2, 5, and 10 neuroinvasive neuroinvasive WND cases were compared to reported case counts using logarithmic [58] and Brier scores [59]. Both the logarithmic and Brier are proper scoring functions that assess probabilistic predictive accuracy. They are scaled such that lower values indicate better predictive performance. Mean Brier scores for the entire US were 0.14, 0.05, 0.02, and 0.01 for the 1, 2, 5, and 10 case count bins, respectively, suggesting improved predictive accuracy with increasing case counts and overall accuracy above 85% at all levels. Logarithmic scores showed similar improvement with increasing case counts having mean scores for the US at 0.53, 0.18, 0.06, and 0.02 for 1, 2, 5, and 10 cases, respectively. Although validation mean scoring for the whole US was suggestive of high predictive accuracy, predictive error was not evenly distributed across the study area. Figure A2 (Appendix A) maps logarithmic scores to illustrate how predictive error varied geographically at each of the case count thresholds.

## 4. Discussion

Perhaps the most striking result of our analysis is the geographic delineation of a region with markedly elevated Relative Risk (Figure 2). As shown in Figure 2 (bottom), a distinct area of elevated neuroinvasive WND risk extends southward from the US–Canada border in the states of Montana, North Dakota, and Wisconsin to the US–Mexico border in western Texas. The elevated risk area runs parallel to the Continental Divide, a predominately mountainous geologic feature that divides US hydrologic watersheds flowing to the Pacific and Atlantic Oceans, and extends eastward through the Great Plains Palouse Dry Steppe Province ecoregion. The Great Plains Palouse Dry Steppe Province falls within the rain shadow of the Rocky Mountains [60]. The overall shape of the elevated disease risk area resembles an inverted triangle effectively dividing the US east-to-west at the Continental Divide. Although the specific factors causing the “risk triangle” pattern are unclear, we speculate that the Rocky Mountains may act as a physical barrier to mosquito and avian species giving rise to a unique vector and host community assemblage that facilitates WNV transmission immediately east of the Rocky Mountains. This speculation is made recognizing that the Continental Divide has previously been identified as a barrier to *Culex tarsalis* and *Culex pipiens* mosquitoes from the plains [61,62] and is known to shape current avian population structure as well as that in the evolutionary past [63,64]. For example, several Passerine species can be genetically differentiated into eastern and western US populations due to decreased gene flow over the Rocky Mountains [65,66]. Human socioeconomic influences may also play a part in elevated risk through this region, however we note that the risk triangle does not appear to coincide or track political boundaries (e.g., US States). Although additional research is required to investigate the risk triangle pattern, one thing is clear, the region exhibits more than twice the neuroinvasive WND risk than it should based on population size.

With respect to climate variables evaluated in our study, precipitation was not found to be an important predictor of disease risk based on its credible interval including zero (Table 2). This is not especially surprising given that our study encompassed the entire US, which exhibits considerable geographic variability in total precipitation over its area. It is also the case that precipitation can have varied effects on vector populations. In some instances, an abundance of precipitation may facilitate mosquito reproduction through the creation of breeding sites, but in other cases elevated rainfall may flush mosquitoes from breeding sites thereby reducing reproduction [67,68]. Precipitation effects on mosquito productivity vary by species, timing, and location with noted disparities between *Aedes* and *Culex* genera, monthly versus annual rain patterns, and differences between the Eastern and Western US [67].

Consistent with prior studies, our analysis indicated an approximate 17.35% increase in the median West Nile case rate for about every 3 ${}^{\circ }$C increase in average maximum temperature. Although this is an important finding, additional work is needed to better link ambient temperature to specific facets of vector physiology [3], behavior [12], overwintering capacity [69], and abundance [70,71]. Despite the large variability in temperature across the country, this variable was positively associated with WND. Temperature has been recognized as playing a major role in shaping the seasonal distribution and timing of WND outbreaks in the US [12,13]. Temperature has also been shown to correlate with increased human disease incidence [15].

Our results suggest that for each increase of 90 WNV positive mosquitoes detected through surveillance, median disease rates increase by approximately 4.08% (Table 2). By comparison, median disease rates increase by about a 3.05% for every 30 birds confirmed to carry WNV. Numerous studies have reported a relationship between mosquito abundance and human West Nile incidence [72,73]. Because mosquitoes transmit virus from avian hosts to humans, they are the primary mediators of the West Nile enzootic cycle, thus, monitoring their abundance and virus prevalence are key indicators of disease risk [11]. It should be noted that mosquito and avian surveillance reporting requirements vary by location, and that considerable differences in surveillance pressure exist at the state, county, and local levels [74].

We found that for every one bird increase in competent avian host richness, median disease rates decreased by 5.13%. Birds are the primary amplifying hosts in the West Nile enzootic cycle [75], however, host competence varies considerably by species making avian community composition a fundamental aspect of West Nile ecology [14]. Indeed, avian community composition in conjunction with mosquito host utilization largely shape disease transmission dynamics [3]. For example, bird community composition and relative species abundances at a location may dictate both the number of overall vector-hosts contacts and the proportion of contacts between vectors and competent bird species. The scenario in which increased host diversity is hypothesized to reduce vector-host virus transmission rates or disease incidence has been referred to as the “dilution effect” [76]. Though our findings suggest some support for a dilution hypothesis, an alternative explanation for this result is that competition and vector feeding preferences decreased transmission rates to humans [77,78,79]. Additional research is necessary to better understand mechanistic connections to avian host richness and to verify that richness is truly associated with decreased disease incidence and not a statistical artefact, or better explained by climate or seasonal changes that birds coincidentally track as part of migration or other facet of their natural history. As previously described for mosquitoes, climate factors like temperature also affect bird physiology and behavior, to include migration, reproduction, and community characteristics [15,80,81].

Changes to vector and host abundances and community composition attributable to temperature variation may aid in explaining intra-annual shifts in disease prevalence, like the onset timing of the summer West Nile outbreak season [12]. However, temperature does not exhibit sufficient irregularity to elucidate the substantial inter-annual fluxes observed in disease risk (Figure 1). As suggested by [82], although temperature is correlated with increased disease incidence, factors related to immunity may drive the epidemic periodicity observed between years in many disease systems. In the interim following a disease outbreak, recently exposed populations may be temporarily protected or buffered from recurrent infection, however as time proceeds, acquired immunity benefits diminish eventually resulting in predominantly naive populations once again vulnerable to epidemic [13,30]. As populations fluctuate between intermittent resistant and naive phases, multiyear epidemic cycles like those illustrated in Figure 1 are observed. The Historic Prevalence variable incorporated into our analysis served as a synthetic immunity index and was scaled to represent the recency and intensity of West Nile exposure in counties. The Historic Prevalence coefficient (Table 2) indicates that for every increase of 0.10 in the index, median neuroinvasive disease rates decreased by approximately 9.42%. That is, locations in our analysis with a relatively high Historic Prevalence recently experienced higher disease incidence and are therefore estimated to have lower disease rates in the immediate future. By comparison, locations with a low Historic Prevalence are analogous to naive populations and therefore anticipated to exhibit greater vulnerability to outbreaks in the near term. Clearly, just as enzootic components of WND are molded by both individual competence and external environmental influences, so too is disease ecology shaped by intrinsic (immunity) and extrinsic (demographic) human factors.

Extrinsic human demographic and economic factors assessed in this study included the proportion of the county population over 54 years of age, total AIANNH population, and median household income (Table 2). We found the population proportion over 54 years to be the strongest risk factor in our analysis. For each 6.41% increase in the population proportion, median neuroinvasive disease risk increased 305.52%. This result is consistent with other research showing a 16 fold increase in risk between 24 and 65 years of age [17], a rate of 1 in 54 for persons 65 years and older [7], and similar elevated risk associated with the aging population [8,16]. Importantly, disease occurrence records used in our analysis did not include any structured demographic information detailing disease cases by age or sex groups; rather, we identified high risk populations through comparison of disease incidence with publicly accessible population estimates provided by the US Census Bureau. In contrast to older populations, AIANNH populations were negatively associated with disease risk such that as the AIANNH population increases, disease risk decreases. However, caution is urged in interpreting this finding because AIANNH demographic structure and average life spans may differ from non-AIANNH populations and disease cases affecting AIANNH populations may be under reported [27]. Median Household Income followed a similar pattern with each $12,271 income increase corresponding to about a 8.33% decrease in the median disease rate. Geographic Area (Table 2), which quantifies variation associated with individual county areal extents (spatial sample sizes), indicated that WND cases increase for counties with areas above the nationwide mean area (approximately 3000 km${}^{2}$). This suggests that additional risk factors need to be considered to fully explain WND occurrence. Despite the need for additional research, Figure 1 and Figure 2 illustrate that interaction between epidemiological, ecological, and socioeconomic processes considered in our analysis culminate in a highly heterogeneous spatial and temporal neuroinvasive WND occurrence pattern.

We applied Bayesian spatiotemporal modeling to assess the association of vector surveillance, host species richness, and a variety of other potential disease risk factors with neuroinvasive WND incidence throughout the conterminous US. Beyond the inherent complexity of quantifying multi-organism disease systems, variation in human directed surveillance effort, sampling protocols, and reporting all complicate and bias epidemiological analysis [18]. Because of the potential for these confounding issues to complicate inference, it is critically important to thoroughly account for imperfect, incomplete, and biased observational data to ensure confidence in findings [83,84]. Key to our approach was joint modeling reported neuroinvasive disease cases with those reported for non-neuroinvasive WND. Modeling both diseases concurrently enabled information from documented non-neuroinvasive disease cases to be leveraged towards improving neuroinvasive disease estimates. Through model comparison, selection, and validation we demonstrated that our chosen methodology produced more accurate and parsimonious results than several alternative approaches. Joint disease modeling in combination with spatially and temporally explicit techniques to account for structured data provided enhanced assurance that revealed relationships between our assessed risk factors and neuroinvasive disease were epidemiological and ecologically relevant and not artifactual. To this end, our research identified spatial misalignment between areas exhibiting highest Relative Risk and those locations experiencing the highest disease caseloads. Our analysis also quantified several important indicators of neuroinvasive disease risk, including factors tied to climatic, demographic, and organismal components of the WND system.

## 5. Summary and Conclusions

Identifying the ecological drivers of neuroinvasive WND and geographically locating those areas most susceptible to elevated disease risk are essential steps in creating an effective disease surveillance, monitoring, and response strategy. To better anticipate West Nile epidemics and elucidate key indicators of risk, we assessed the importance of several climatic, demographic, economic, and environmental factors. Central to our methodology was integrated modeling of neuroinvasive cases with those reported for non-neuroinvasive WND. By evaluating both diseases concurrently information describing the spatiotemporal pattern of non-neuroinvasive disease was used to improve the accuracy of neuroinvasive disease estimates. Our research revealed that human demographic factors connected to aging populations were the strongest disease indicators, but extrinsic climatic and vector-host biotic interactions need also be considered when appraising West Nile risk. Our analysis also identified a region of disproportionately high neuroinvasive WND disease paralleling the Continental Divide between Canada and Mexico. Our results can be applied to identify locations for priority disease surveillance and we hope that the described approach will motivate future research into modeling disparate facets of disease systems at the wildlife-human interface.

## Figures and Tables

**Figure 1 viruses-13-00934-f001:**
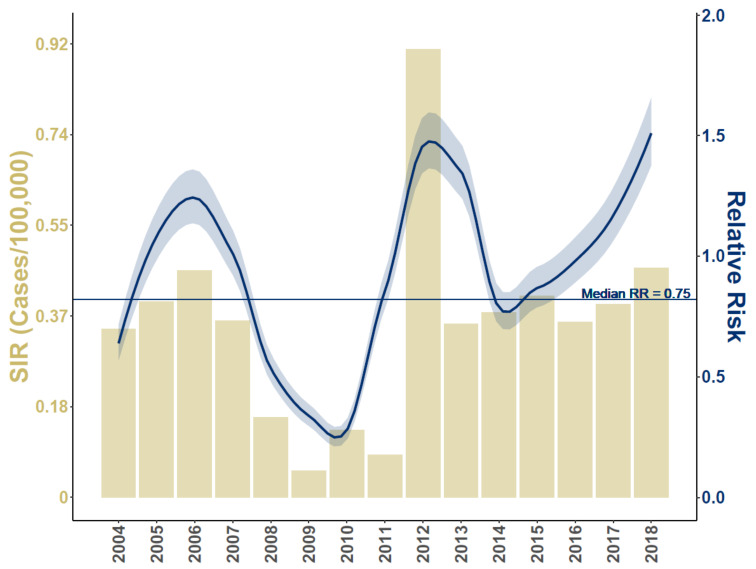
Average Standardized Incidence Rates (SIR) and Relative Risk for period of record. Horizontal axis lists time (year), left vertical axis corresponds to histogram and describes the SIR (neuroinvasive Cases/100,000 people), and right vertical axis relates to curvlinear line and indicates estimated Relative Risk with respect to 2004–2018 median risk (straight, horizontal line at Relative Risk RR = 0.75). Shaded area around curvlinear line provides the 95% credible interval. Relative Risk is the ratio of model estimated disease cases to the expected case number given the population size. A Relative Risk value of 1 indicates that model predicted cases were comparable to the expectation, values below 1 indicate periods of relatively low risk, and values above 1 suggest increased risk.

**Figure 2 viruses-13-00934-f002:**
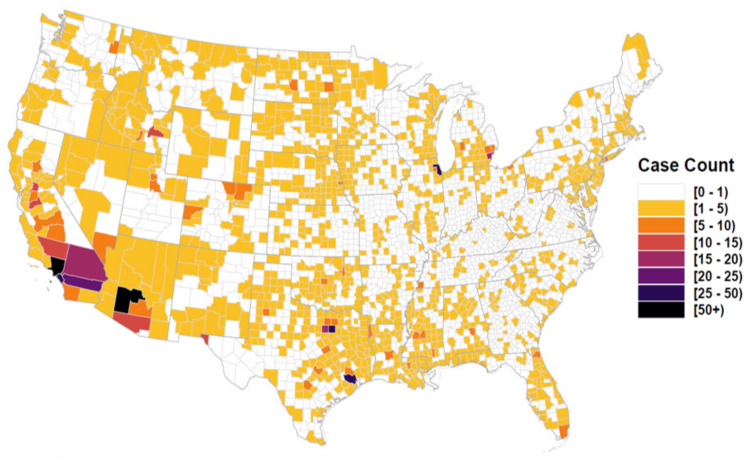
Estimated case counts and Relative Risk. Figure displays the estimated neuroinvasive disease case number (**top**) and Relative Risk (**bottom**) for US counties. Figures are color coded such that darker colors indicate more disease cases and higher Relative Risk. Map areas shown as white symbolize estimated values near zero. Relative Risk is the ratio of model estimated disease cases to the expected case number given the population size. A Relative Risk value of 1 indicates that model predicted cases were comparable to the expectation, values below 1 indicate locations of relatively low risk, and values above 1 suggest increased risk. Note that elevated Relative Risk areas do not always coincide with locations experiencing highest case counts.

**Figure 3 viruses-13-00934-f003:**
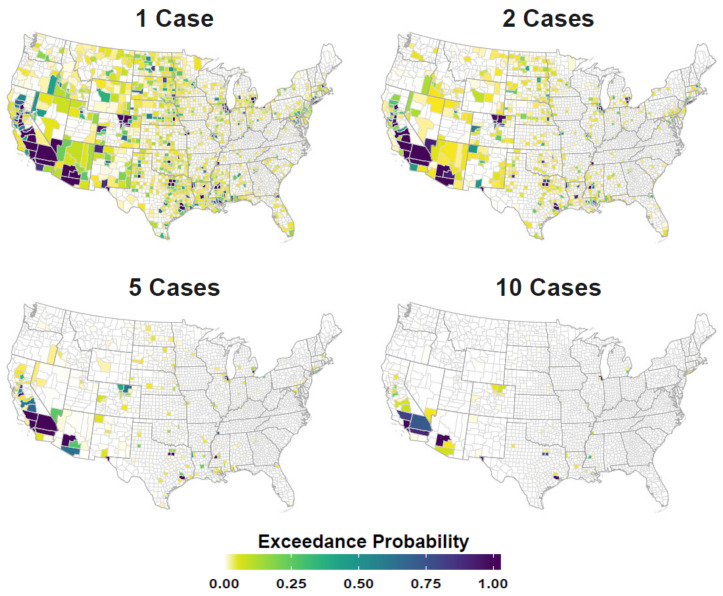
Case exceedance probability. Figure displays the probability of counties having more than 1, 2, 5, and 10 neuroinvasive disease cases annually. Maps are color coded such that darker tones (deep blues) indicate higher exceedance probabilities (0.00–1.00) and lighter colors (greens and yellows) represent relatively lower probabilities.

**Figure 4 viruses-13-00934-f004:**
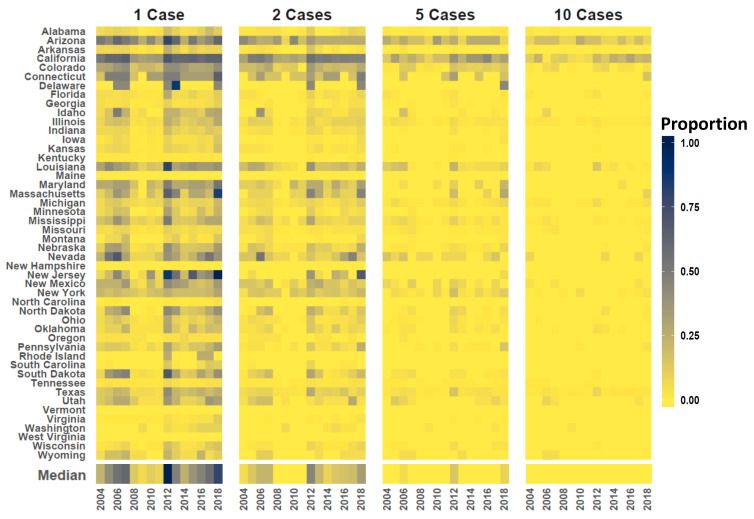
State case counts. Vertical axis at left lists US State names (aligned as rows) and horizontal axis shows time (year). Color code indicates the proportion counties in each state estimated to experience 1, 2, 5, and 10 cases. Dedicated panels are provided from left to right for 1, 2, 5, and 10 cases, respectively. Darker tones indicate a relatively high proportion of counties in the state are subject to the given case count and lighter tones signify a lower proportion. Color coded bar parallel to horizontal axis gives the median proportion of US Counties subject to each case threshold.

**Table 1 viruses-13-00934-t001:** Model comparison. Deviance information criterion (DIC) and Watanabe–Akaike information criterion (WAIC) for disease models. Lower values indicate improved parsimony. Full, joint disease model (Model5) exhibited the best parsimony.

Model	DIC	WAIC	Description
Model1	65491	65690	Non-spatiotemporal (All fixed covariates)
Model2	40699	40355	Spatiotemporal (No fixed covariates)
Model3	40281	39798	Individual Neuroinvasive (All covariates)
Model4	55937	55889	Joint Disease (No fixed effects)
Model5	38680	38082	Full Joint Disease (All covariates)

**Table 2 viruses-13-00934-t002:** Estimated fixed effect coefficients. Model estimated coefficients for West Nile neuroinvasive disease. Mean, standard deviation (SD) and 95% Credible Interval as estimated by the joint disease model (Model5). Coefficients are on the log scale.

Covariate	Mean	SD	2.5 Q	97.5 Q
Intercept	−0.54	0.17	−0.86	−0.22
Median Household Income	−0.08	0.02	−0.12	−0.04
Historic Prevalence	−0.09	0.02	−0.12	−0.05
Proportion ≥54 Years	1.40	0.08	1.25	1.55
County Geographic Area	0.71	0.27	0.18	1.24
Competent Host Richness	−0.05	0.01	−0.06	−0.05
Max Temperature	0.16	0.04	0.09	0.23
Total Precipitation	0.03	0.02	−0.01	0.07
WNV Mosquito Detection	0.04	0.01	0.04	0.05
WNV Avian Detection	0.03	0.01	0.02	0.04
AIANNH Population	−0.04	0.01	−0.06	−0.01
AIANNH Lands	0.07	0.01	0.04	0.09
Disease Interaction (λ)	0.89	0.20	0.84	0.92

## Data Availability

Data used in this study are freely available and can be obtained from the Centers for Disease Control and Prevention [4].

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
