# Peer review of "Vector Surveillance, Host Species Richness, and Demographic Factors as West Nile Disease Risk Indicators"

_viruses, 2021, doi:10.3390/v13050934_

Round 1

Reviewer 1 Report

The manuscript represents a comprehensive statistical analysis of collected data that are commonly recognized as risk factors for neuroinvasive West Nile disease (WND) occurrence as well as a number of them that are or could be risk factors specific to US. It is well written, well organized and includes all relevant data. To my opinion the textual part of the manuscript could be a bit more condensed for reader’s convenience, but this does not diminish the value of the manuscript in the current form.

Note that neuroinvasive WND presents aseptic meningitis, and not septic meningitis as stated in the line 28. Please correct.

Author Response

COMMENT: The manuscript represents a comprehensive statistical analysis of collected data that are commonly recognized as risk factors for neuroinvasive West Nile disease (WND) occurrence as well as a number of them that are or could be risk factors specific to US. It is well written, well organized and includes all relevant data. To my opinion the textual part of the manuscript could be a bit more condensed for reader’s convenience, but this does not diminish the value of the manuscript in the current form.

RESPONSE: Thank you for your appreciation of the manuscript.  To better condense the manuscript for readers, the following revisions have been implemented:

  1. Text at Lines 75-79 has been simplified.
  2. Lines 107-112 have been rephrased for ease of interpretation.
  3. Text at Lines 119-121 and Lines 437-441 has been clarified.
  4. Historic Prevalence maps (previous Fig A2) and all associated text have been removed.

As an additional note, although we recognize that the Methods section of the manuscript is quite detailed, we opted not to make additional cuts there because this project was in-part initiated pursuant to the CDC’s West Nile Forecasting 2020 challenge mentioned in the Acknowledgements section.  We anticipate that other researchers involved in the CDC challenge may have interest in the additional statistical detail. 

COMMENT: Note that neuroinvasive WND presents aseptic meningitis, and not septic meningitis as stated in the line 28. Please correct.

RESPONSE: Thank you.  The reviewer is correct and the term “septic” has been revised to “aseptic” when referencing the WND clinical definition (see revised Line 28) .

Reviewer 2 Report

Firstly I would like to compliment the authors for the manuscript and the work done. I think this is a good paper, which merits to be published Although the results are not really adding a lot of new information on WND epidemiology, the methodology presented is interested.

I have only few comments and little concerns.

My first concern is in the use of the so called "Historic Prevalence variable as predictor. In fact, this variable is calculated based on the same variable (human incidence data) used as response variable. This is unusual approach. I understand the scope of this variable but it seems a little bit methodologically not correct to use a variable based on the incidence as a predictor of the expected incidence. I would suggest to modify this variable considering only the frequency of occurrence of the disease in the same county, thus avoiding to consider the quantitative aspect of the incidence. I have also to say that the results regarding this variable are quite contradictory. If I look at the Figure A2 I see that the same areas (more or less) with the higher Historic Prevalence are those with the higher Relative Risk (figure 2). This would contradict the results (Table 2) on which an inverse effect of this variable is expected.

Other minor points:

  • Table 2 shows that "County Geographic Area" has a quite strong effect. But this variable has been poorly explained in the Materials and Methods. I have understood that this is the extension of each county. Is it correct? Please explain more clearly the calculation of this variables and include some comments on its effects in discussion (currently this variable is not mentioned in discussion, albeit its effect is quite strong [0.71]).
  • At lines 74-76, avian and mosquito surveillance data are reported as response data, although they have been used as predictors.
  • Lines 361-368 (Discussion): something should be said about differences in surveillance pressure among States and counties. 
  • Lines 369-378 (Discussion): Another explanation for what observed is based on the fact that a higher avian host richness can reduce the probability of infection passage to human population. In fact, the host competition in a such situation would be favourable for birds species, in respect to humans. Several paper explained this possible mechanism for WND and other infectious diseases with wild reservoirs.
  • Lines 421-425 (Discussion): are the native population on average younger than the other US populations? If the life expectancy is lesser, this could explain the negative association with risk.

Author Response

COMMENT: Firstly I would like to compliment the authors for the manuscript and the work done. I think this is a good paper, which merits to be published Although the results are not really adding a lot of new information on WND epidemiology, the methodology presented is interested.

RESPONSE: Thank you for the compliment and we appreciate the reviewer’s comments and suggestions.  Please see below for comment-specific responses.

COMMENT: I have only few comments and little concerns.  My first concern is in the use of the so called "Historic Prevalence variable as predictor. In fact, this variable is calculated based on the same variable (human incidence data) used as response variable. This is unusual approach. I understand the scope of this variable but it seems a little bit methodologically not correct to use a variable based on the incidence as a predictor of the expected incidence. I would suggest to modify this variable considering only the frequency of occurrence of the disease in the same county, thus avoiding to consider the quantitative aspect of the incidence.

RESPONSE:  We concur that use of the “Historic Prevalence” variable is not typical and recognize that because of this its associated textual description was incomplete.  We have revised the manuscript to better clarify use of the Historic Prevalence predictor as well as its relationship to the more commonly used variable and term “Force of Infection”.  To clarify here, we used Historic Prevalence in an analogous fashion to Force of Infection (FOI) in the sense that both refer to the per capita rate at which susceptible individuals in a population acquire an infection.  As traditionally applied, FOI is estimated from apparent (sampled) or calculated cumulative incidence to approximate age-related changes in disease incidence (Grenfell and Anderson, 1985).  However, because data detailing human population age structure and individual infection status are rare at large spatial extents, many epidemiological studies have proposed departing from traditional usage to approximate FOI dynamics from historic case counts, vector densities, or past outbreaks (Medone et al., 2015; Manrique et al., 2016; Sallam et al., 2017).  Inclusion of FOI is most common in mathematical epidemiology (compartmental models and simulation-based studies), but has also been explored to a lesser extent in mixed models (Braga et al., 2010; Hamer et al., 2011), including those for West Nile (Levine et al., 2016).  In the current study, we approximated infection rates from total population size and historic outbreaks (Massad et al., 2016), but chose not to refer to this predictor as a FOI to avoid confusion with the term’s original usage (sensu Grenfell and Anderson, 1985) and those epidemiological studies that continue to incorporate and interpret FOI in the traditional (strict) manner.

In terms of methodological correctness, Historic Prevalence was calculated as a population-weighted, moving average (scaled, continuous numbers) and then time lagged by 1 year in relation to the response variables.  By comparison, response variables indicate county-level, total annual disease case counts (integers, not a per capita incidence or rate).  That is, Historic Prevalence is a descriptive statistic of past, between-year, and per capita rate changes and does not reflect current, within-year, county-based frequencies as do the response variables.  A major challenge in disease forecasting is accounting for changes in underlying (per capita) immunity due to past infection or vaccination that may not be identified by modeling only case counts (Wearing et al., 2006; Johansson et al., 2011; Johansson et al., 2016; Paull et al., 2017); we argue that variables like the FOI and our Historic Prevalence aid in addressing some of these population intrinsic factors.

Lastly, having time-lagged and evaluated variables for potential multicollinearity (Line 123), there is little concern that Historic Prevalence statistically confounds, overfits, or otherwise conflates other variables.  In support of this point, parsimony metrics (Table 1), credible intervals (Table 2), and validation measures (updated Fig A2) indicate that the Historic Prevalence variable is 1) sufficiently generalized to not confound the response, 2) explains response variance beyond that explained through fitting of annual case counts alone, 3) there is no loss or stagnation between fitted and predicted values (i.e., estimates from training data are not more accurate than those obtained from true prediction ), and 4) inclusion of Historic Prevalence improves overall parsimony.

A much abbreviated version of this explanation has been added at Lines 107-112.

References:

Braga, C., Luna, C. F., Martelli, C. M. T., Souza, W. V. de, Cordeiro, M. T., Alexander, N., Albuquerque, M. de F. P. M. de, Júnior, J. C. S., & Marques, E. T. (2010). Seroprevalence and risk factors for dengue infection in socio-economically distinct areas of Recife, Brazil. Acta Tropica, 113(3), 234–240.

Grenfell, B. T., & Anderson, R. M. (1985). The estimation of age-related rates of infection from case notifications and serological data. The Journal of hygiene, 95(2), 419–436.

Hamer, G. L., Chaves, L. F., Anderson, T. K., Kitron, U. D., Brawn, J. D., Ruiz, M. O., Loss, S. R., Walker, E. D., & Goldberg, T. L. (2011). Fine-scale variation in vector host use and force of infection drive localized patterns of West Nile virus transmission. PLoS ONE, 6(8).

Johansson, M. A., Hombach, J., & Cummings, D. A. (2011). Models of the impact of dengue vaccines: a review of current research and potential approaches. Vaccine, 29(35), 5860–5868.

Johansson, M., Reich, N., Hota, A. et al.(2016). Evaluating the performance of infectious disease forecasts: A comparison of climate-driven and seasonal dengue forecasts for Mexico. Sci Rep 6, 33707.

Levine, R. S., Mead, D. G., Hamer, G. L., Brosi, B. J., Hedeen, D. L., Hedeen, M. W., McMillan, J. R., Bisanzio, D., & Kitron, U. D. (2016). Supersuppression: Reservoir Competency and Timing of Mosquito Host Shifts Combine to Reduce Spillover of West Nile Virus. The American journal of tropical medicine and hygiene, 95(5), 1174–1184.

Manrique, P. D., Xu, C., Hui, P. M., & Johnson, N. F. (2016). Atypical viral dynamics from transport through popular places. Physical review. E, 94(2-1), 022304.

Massad, E., Tan, S. H., Khan, K., & Wilder-Smith, A. (2016). Estimated Zika virus importations to Europe by travellers from Brazil. Global Health Action, 9(1).

Medone, P., Ceccarelli, S., Parham, P. E., Figuera, A., & Rabinovich, J. E. (2015). The impact of climate change on the geographical distribution of two vectors of chagas disease: Implications for the force of infection. Philosophical Transactions of the Royal Society B: Biological Sciences, 370(1665), 1–12.

Paull, S. H., Kilpatrick, A. M., Horton, D. E., Diffenbaugh, N. S., Ashfaq, M., Rastogi, D., & Kramer, L. D. (2017). Drought and immunity determine the intensity of west nile virus epidemics and climate change impacts. Proceedings of the Royal Society B: Biological Sciences, 284(1848).

Wearing, H. J., & Rohani, P. (2006). Ecological and immunological determinants of dengue epidemics. Proceedings of the National Academy of Sciences of the United States of America, 103(31), 11802–11807.

COMMENT: I have also to say that the results regarding this variable are quite contradictory. If I look at the Figure A2 I see that the same areas (more or less) with the higher Historic Prevalence are those with the higher Relative Risk (figure 2). This would contradict the results (Table 2) on which an inverse effect of this variable is expected.

RESPONSE: Thank you for highlighting the lack of clarity in (original) Fig A2.  This figure and associated text have been removed to simplify the overall manuscript and to avoid potential misinterpretation.  To address the apparent contradiction issue, (original) Fig A2 was not intended as a comparison between different counties or different locations, rather it was meant to illustrate changes through time within single (individual) counties.  For example, it would not be contrary for a location to have a negative Historic Prevalence coefficient (Table 2) provided that case counts decrease as Historic Prevalence increases through time within that individual county.  This negative relationship would hold true even if that individual county consistently exhibited both higher Historic Prevalence and higher risk than all other counties in the US. 

COMMENT: Other minor points:  Table 2 shows that "County Geographic Area" has a quite strong effect. But this variable has been poorly explained in the Materials and Methods. I have understood that this is the extension of each county. Is it correct? Please explain more clearly the calculation of this variables and include some comments on its effects in discussion (currently this variable is not mentioned in discussion, albeit its effect is quite strong [0.71]).

RESPONSE:  Thank you for identifying this oversight.  The manuscript has been revised to clarify use and interpretation of the County Geographic Area variable.  Please see revised Lines 119-121 and Lines 437-441.  To clarify here, geographic area (the areal extent or size of a spatial unit) is a common variable to include in ecological and spatial modeling under the probabilistic assumption that the chance of a random event occurring (or being observed) increases with sample unit size. 

COMMENT: At lines 74-76, avian and mosquito surveillance data are reported as response data, although they have been used as predictors.

RESPONSE:  Thank you.  Description of avian and mosquito surveillance data have been relocated from the response data section to the “driver” section as suggested (Lines 75-77).

COMMENT: Lines 361-368 (Discussion): something should be said about differences in surveillance pressure among States and counties.

RESPONSE:  Thank you, authors fully concur with the reviewer and have revised Lines 373-375 to note that considerable variation in surveillance pressure exist at the state, county, and local levels with a link provided to the CDC WNV Surveillance website reporting the same.

COMMENT: Lines 369-378 (Discussion): Another explanation for what observed is based on the fact that a higher avian host richness can reduce the probability of infection passage to human population. In fact, the host competition in a such situation would be favorable for birds species, in respect to humans. Several paper explained this possible mechanism for WND and other infectious diseases with wild reservoirs.

RESPONSE:  The authors agree with the reviewer’s alternative explanation and have revised the manuscript to indicate that alternative explanations to the dilution hypothesis include the possibility that competition and vector feeding preferences contributed to decreased transmission rates to humans (Hamer et al., 2011; Simpson et al., 2012; Marini et al., 2017).  Please see Lines 386-390.

References:

Hamer, G. L., Chaves, L. F., Anderson, T. K., Kitron, U. D., Brawn, J. D., Ruiz, M. O., Loss, S. R., Walker, E. D., & Goldberg, T. L. (2011). Fine-scale variation in vector host use and force of infection drive localized patterns of West Nile virus transmission. PLoS ONE, 6(8).

Marini, G., Rosá, R., Pugliese, A., & Heesterbeek, H. (2017). Exploring vector-borne infection ecology in multi-host communities: A case study of West Nile virus. Journal of Theoretical Biology, 415(January 2016), 58–69.

Simpson, J. E., Hurtado, P. J., Medlock, J., Molaei, G., Andreadis, T. G., Galvani, A. P., & Diuk-Wasser, M. A. (2012). Vector host-feeding preferences drive transmission of multi-host pathogens: West Nile virus as a model system. Proceedings of the Royal Society B: Biological Sciences, 279(1730), 925–933.

COMMENT: Lines 421-425 (Discussion): are the native population on average younger than the other US populations? If the life expectancy is lesser, this could explain the negative association with risk.

RESPONSE:  Thank you, the reviewer’s point is valid and an important consideration.  Although the native population data used in this study is insufficient to make a rigorous demographic comparison to non-native populations, the manuscript has been revised to caveat that (in addition to reporting bias for native populations) native demographic structure and average life spans may differ from those in the non-native population (Lines 432-436).